# Species-specific thermal classification schemes can improve climate related marine resource decisions

R. Freedman[1,2ɵ]*, J. A. Brown[1,3ɵ], C. Caldow[1ɵ], J. E. Caselle[4ɵ]

**1** NOAA Channel Islands National Marine Sanctuary, Santa Barbara, CA, United States of America,
**2** Ecology Evolution and Marine Biology Department, University of California Santa Barbara, Santa Barbara, CA, United States of America, **3** ECOS Consulting, LLC, Lafayette, CA, United States of America, **4** Marine Science Institute, University of California Santa Barbara, Santa Barbara, CA, United States of America

ɵ These authors contributed equally to this work.
* ryan.m.freedman@noaa.gov

**Data Availability Statement:** Data used is publicly available as described in the manuscript. VertNet (http://vertnet.org) and Fishbase (https://www.

## Abstract

Global climate change increasingly contributes to large changes in ecosystem structure. Timely management of rapidly changing marine ecosystems must be matched with methods to rapidly quantify and assess climate driven impacts to ecological communities. Here we create a species-specific, classification system for fish thermal affinities, using three quantifiable datasets and expert opinion. Multiple sources of information limit potential data bias and avoid misclassification. Using a temperate kelp forest fish community in California, USA as a test case for this new methodology, we found the majority of species had high classification agreement across all four data sources (n = 78) but also a number of low agreement species (2 sources disagree from the others, n = 47). For species with low agreement, use of just one dataset to classify species, as is commonly done, would lead to high risk of misclassification. Differences in species classification between individual datasets and our composite classification were apparent. Applying different thermal classifications, lead to different conclusions when quantifying 'warm' and 'cool' species density responses to a marine heatwave. Managers can use this classification approach as a tool to generate accurate, timely and simple information for resource management.

## Introduction

Climate change is one of the greatest challenges natural resources will face over the next century. While species are generally predicted to shift their ranges in response to warming conditions, individual ecosystems will have unique responses depending on local environmental variables and species' sensitivities to environmental conditions [1–3]. As species disappear from or arrive into ecosystems as a function of shifting climate, the ecosystem services provided will likely be altered and managers must develop adaptive strategies to meet management goals (e.g., optimal take for stocks, MPA placement for conservation; [1, 4–6]). In order for resource management to be timely and responsive to climate changes, effective

fishbase.de) are publicly available online. PISCO data can be found at DataOne (https://www.dataone.org). Kelp Forest Surveys: doi:10.6085/AA/PISCO_kelpforest.1.6. and Subtidal Recruitment Data https://doi.org/10.6085/AA/PISCO_UCSB_Fish_Recruitment.1.3.

**Funding:** The National Oceanic and Atmospheric Administration (NOAA) provided support in the form of salaries for authors associated with ECOS Consulting, namely Dr. Jennifer Brown. NOAA's Integrated Ecosystem Assessment program provided funding to support this work. The funders had no role in study design, data collection and analysis, decision to publish, or preparation of the manuscript. NOAA's support for ECOS was to allow Dr. Brown to work with other NOAA authors and participate in a fair and unbiased role in the study design, data collection and analysis, decision to publish, or preparation of the manuscript. ECOS consulting did not play a role in study design, data collection and analysis, decision to publish, or preparation of the manuscript. PISCO is funded in part by the David and Lucille Packard Foundation. This funding support does not alter our adherence to PLOS ONE policies on sharing data and materials.

**Competing interests:** ECOS consulting did not have any additional role in the study design, data collection and analysis, decision to publish, or preparation of the manuscript. Jennifer Brown's commercial affiliation with ECOS consulting does not alter our adherence to PLOS ONE policies on sharing data and materials." (as detailed online in our guide for authors http://journals.plos.org/plosone/s/competing-interests). The specific roles of these authors are articulated in the 'author contributions' section.

methodologies must be available to track changes in ecosystem components, such as community structure or function.

Classifying species by their affinities for environmental conditions can improve our ability to understand how local ecological communities are changing in response to acute and chronic climatic events and to contextualize any changes relative to specific environmental drivers. Because physiological data on most species' thermal tolerances is absent, the most commonly used classifications are based on species thermal tolerances as inferred from biogeography *in situ* [4, 7, 8]. However, some classification schemes use a variety of data metrics including life history traits [9], other (non-thermal) environmental sensitivities and even co-occurring human activities [10]. Classification schemes create using readily available datasets, such as abundance or distribution monitoring data, can facilitate rapid information generation and quick management decision-making. Species classification schemes that are easily understood will best facilitate management communications with stakeholders and ultimately management action [11].

The most useful classification scheme for management uptake will not only be intuitive but also reliable. Reliability is dependent on the classification accuracy and sensitivity to detect changes that resource management would be concerned about. Classification schemes with high levels of erroneous classification could lead to overreaction or inaction. The risk of error and bias is especially high for schemes that use just a single data source to classify species or those that relay information in a single, composite indicator, such as the commonly employed Community Thermal Indicators (CTI; [7, 8, 12]). To alleviate spatial and temporal shortcomings of individual, often site-based or single time point data sources, researchers should utilize multiple data sources to build thermal classification schemes, including the expert opinion of local researchers, traditional resource users and naturalists.

In order to help regional resource managers understand climate impacts, we developed a robust thermal classification for a kelp forest fish community off the coast of California, USA. We created a composite classification using multiple sources of information and investigated agreement between the results of each individual source. Fishes were sorted into warm-affinity and cool-affinity (hereafter 'warm' and 'cool') species classifications using information from four different data sources (i.e., Expert Opinion, Range Limits, Museum Collections, and In Situ Density, see methods for details). Species were classified using each data source separately, and then a final aggregate classification was given to each species based on the majority agreement of classifications across the four data sources. An agreement score between the classification based on the four different sources was quantified to assess potential uncertainty in the classification scheme. The classification scheme was then used to evaluate changes in fish communities in the Santa Barbara channel before and after a marine heatwave event.

## Materials and methods

### Site

The Santa Barbara Channel is a marine thermal transition zone between the cooler California Current and the warmer California Countercurrent (See S1 Fig). The region extends up to Point Conception, a well-established biogeographic break along the Northeast Pacific. Given that this area is a marine transition zone, we expect that fish species will be at the edge of their physiological thermal width [13] and may be able to quickly respond to climate impacts. The mainland coast is primarily south-facing with a series of nearshore kelp forest and rocky reefs. Four islands with kelp forest and rocky reef habitat are situated offshore, separated from the mainland by a deep basin. The Santa Barbara Channel waters are managed through a number of state (e.g., California Fish and Wildlife, California State Lands Commission, California State

Water Resources Control Board) and federal (e.g., National Oceanic and Atmospheric Administration, National Park Service) agencies with many mechanisms of conservation including no take and limited take Marine Protected Areas (hereafter referred to as MPAs).

## Kelp forest monitoring data

Dive surveys (<70 ft) have been conducted in the Santa Barbara Channel annually beginning in 1999 and continuing through present. The fish species list (n = 134) for this study was generated from all species counted in these surveys. The data used in this study were from a long-term dataset collected from 59 sites that were sampled annually by the Partnership for Interdisciplinary Studies Coastal Oceans (PISCO) from June to October; however, not all sites were surveyed in all years. At each study site divers conducted 8 to 12 transects that were 30x2x2m at each of three levels in the water column: benthic, midwater and kelp canopy (when the canopy was present at a site). Transect locations were selected through a stratified random design with multiple non-permanent transects located in fixed strata (e.g., outer, middle, and inner rocky reef). On each transect, a single SCUBA diver counted and estimated the total length in centimeters for each fish, excluding small cryptic fishes. Full techniques for subtidal dive surveys can be found online at www.piscoweb.org.

PISCO also measures recruitment using artificial larval fish collectors known as Standard Monitoring Units for Recruitment of Fishes (SMURFs) [14]. At each of seven sites in the Santa Barbara Channel, three replicate SMURFs were sampled bi-weekly and each fish recruit was identified to the lowest taxonomic level possible [15]. An additional 18 species detected in SMURF samples, but not observed during PISCO dive surveys, were considered and classified as part of this effort, bringing the total number of species included in this study to 152.

## Classification scheme

Each fish species was assigned a thermal classification of warm or cool based on the species' biogeographic distribution patterns relative to Point Conception. Three sources of quantitative data were compiled: *in situ* densities, museum/aquarium collection events, and geographic range midpoints. A fourth, categorical data source, was collected using an expert opinion poll. Expert opinions are an additional source of information that can be incorporated without the limitations of missing data or potential bias from artificially extended range limits. Final classification of a species to warm or cold thermal groups was based simply on the majority case for each of these four data sources.

1. Density data used for classification were from a California wide SCUBA survey using methods described above. We used data from 2009–2010, two years where all sites across the entire California Coast were sampled. While surveys were conducted as part of the same monitoring program (i.e., PISCO), this larger scale snapshot does not overlap the timeframe of the data used for the community change analysis described below and therefore should not be influenced by the 2014 marine heatwave or lead to redundancies in the analysis. If the density for any given species was greater above Point Conception (see S1 Fig), the species was classified as cool; if lower the species was classified as warm. If a species was not present in the data, then it was unclassified.

2. Locations of species from museum and aquarium collections were gathered from Vertnet. org, a public online database. Vertnet has collection data from many museums and aquaria worldwide; the major collecting groups in the California Current region are Birch Aquarium, California Academy of Sciences, Aquarium of the Pacific, Seattle Aquarium, Monterey Bay Aquarium, Natural History Museum of Los Angeles. These collections span a long time

period and allow this dataset to capture spatial data across a variety of climatic conditions in the region. The number of collection events north and south of Point Conception were counted and compared for each species to assign a thermal classification.

3. Range limit data were collected from www.FishBase.org; field guides were used when range limits were not available in Fishbase. The midpoint was calculated to be the average of the upper and lower range limit's latitude. Species with midpoints above Point Conception were classified as cool while those below Point Conception were classified as warm. If counts were equal above or below Point Conception for spatial data sources, the species was considered eurythermal. Eurythermal species were not used in further analysis. If data was not available to determine a classification, the species was deemed unclassified for that data source.

4. In addition to sources described above, we collected categorical data using two online expert opinion polls that were distributed to ichthyologists along the U.S. West Coast at a number of academic institutions (i.e., University of California Santa Barbara, University of California Santa Cruz, California State University Northridge, California State University Long Beach and Scripps Institute of Oceanography) and a professional society, the Southern California Association of Ichthyological Taxonomists and Ecologists (SCAITE). The first survey focused on species recorded in subtidal diver surveys and we received seven completed surveys from fish biologists across four institutions. The second follow-up survey, which focused on species counted in SMURFs but not present in subtidal dive surveys, had four participants from SCAITE. Online expert survey participants were asked to select one option from six available to categorize each species: Temperate, Sub-Temperate, Central, Sub-Tropic, Tropic and Cosmopolitan. Terms were not defined for participants in order to limit bias. Our intent was to gauge the expert's qualitative expectations of species thermal affinities beyond their current and past distributions (e.g., taking into account phylogeny and other characteristics). While we did not explicitly tell experts to not use tools, such as FishBase, to identify species ranges,—they were encouraged to rely on their natural history insight. Post hoc comparisons suggest it is unlikely that experts gathered information from other sources; the number of cool and eurythermal species classified by expert opinion and ranges from Fishbase were the two most dissimilar. For this study, these thermal classifications were then simplified to warm, cool and eurythermal to align expert opinion data to the classification terms assigned using the quantitative data sources: Temperate and Sub-Temperate were assigned cool, Subtropic and Tropic were assigned warm, and the remaining two were assigned eurythermal.

Data from the expert opinion poll and the three sources of biogeographic data were combined to create a single composite thermal classification for each fish species. Here we gave equal weight to each data source and a species was given a final classification of warm, cool, or eurythermal based on the classification most frequently assigned across the four data sources. For species that did not have representation in all data sources (for example not counted on dive surveys), we used any remaining available data to classify them. If classifications from different data sources were evenly split between warm and cool, the final classification was eurythermal.

A classification agreement rank for each species was made to quantify the agreement across data sources. If all four data sources produced the same classification, agreement was considered high. For species with one data source that was in disagreement with the others, the species was considered to have Moderate agreement. A species was considered to have Low agreement when 2 data sources disagreed from the others or no consensus was apparent.

## Community response comparisons

In 2014, the Santa Barbara Channel experienced a marine heatwave that led to anomalously high-water temperatures that persisted until 2016 [16]. Using the marine heatwave as an *in situ* experiment to compare the performance of composite classification with the classifications resulting from single data sources, we truncated the subtidal survey data to the years immediately before ('pre-heatwave'; 2012–2013) and during ('heatwave'; 2014–2015) the marine heatwave. The warm and cool fish densities from subtidal diver surveys were averaged across survey sites in the Channel Islands and sums of site averages were plotted as percent change for the pre-heatwave and heatwave timeframes for each thermal group. Pre- and post-heatwave densities of individual species were plotted against one another, for each classification metric (including the composite metric) separately, and qualitatively described. Density metrics were selected to test the thermal classification for any response to the heatwave because density measures will be inclusive of a number population and demographic responses including larval recruitment, increased survivorship adult migration and increased habitat use.

## Results

In the composite classification of all species (n = 152), 59 species (38.8%) were considered warm and 74 species (48.7%) were considered cool (S1 Table). Across each classification data source, the number of species falling in each group varied greatly: 44–59 for warm species, 32–83 for cool species and 1–29 for eurythermal species. Each of the data sources had a number of unclassified species; however, the 'diver surveys' was the data source with the highest number of unclassified species (n = 61).

We ranked the level of agreement of thermal affinity classifications (warm, cold, eurythermal) for each species across the four different data sources (i.e., Expert Opinion, Range Limits, Museum Collections, and Diver Surveys). While the majority of species had High classification agreement across all four data sources (n = 78, 51%), some species had Moderate (1 data source is in disagreement from the others, n = 27, 18%) or Low agreement (2 sources disagree from the others, n = 47, 31%).

All data sources captured the increase in warm species and the decrease in cool species after the onset of the marine heatwave except for range limits, which did not show a decrease in cool species (Fig 1). Warm species tended to show larger percent changes compared to cool species. Dive Surveys was also the data source with the smallest increase in density of warm water species (127.07% increase in density) but also showed the largest decrease in cool water species (47.34% decrease in density). Museum Collections showed the largest increase in warm water species (134.38% increase in density). The composite score was in the middle of the recorded changes for the four data sources both warm and cool species. In the ten species that had the largest differences pre and post heatwave (Fig 2), six were of high agreement, two were moderate agreement and two were low agreement (Fig 2). These strongest species responses were split pretty evenly between warm (n = 6) and cool species (n = 6). The summed changes in fish density responses to the heatwave were heavily driven by these top 10 species which was 6 times total observed change in density for the remaining species (Fig 2).

## Discussion

Species classifications based on traits that are responsive to climate change (e.g., thermal and otherwise) may be prone to biases depending on the classification method employed and these biases can lead to inconsistencies among studies [8, 11, 17, 18]. Species responses to temperature shifts will vary both on temperature tolerance width, maximum survivable temperature, preferred temperature and the plasticity around these traits [4, 6]. In our system, thermal

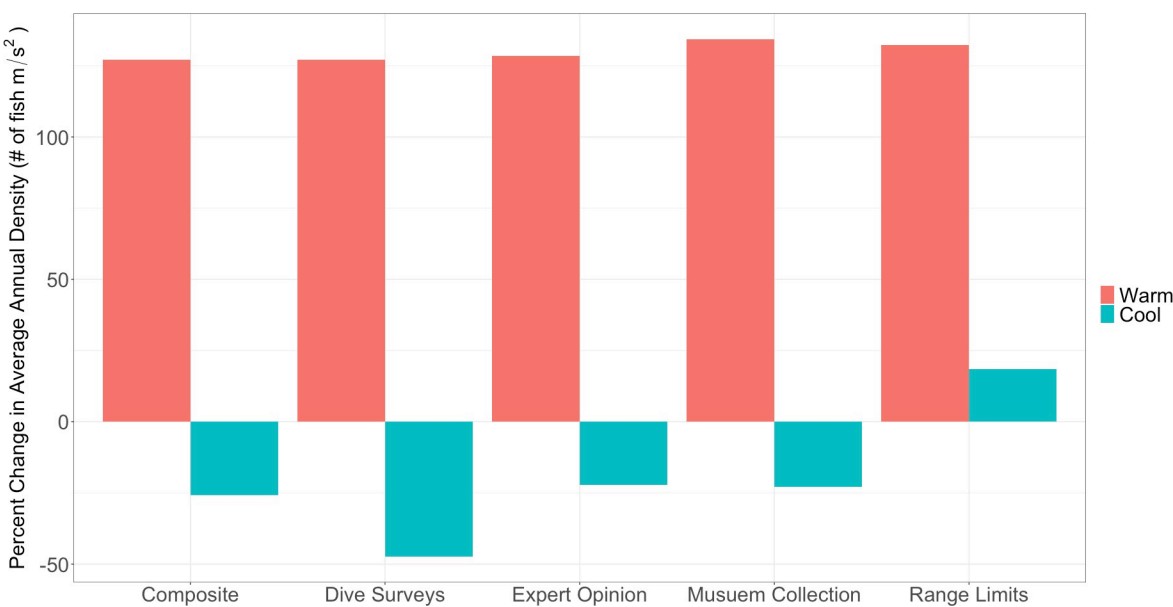

**Fig 1. Changes in average annual fish densities.** Percent changes in average annual fish densities for warm (red bars) and cool (blue bars) species are shown for each classification data source for the years prior to (2012–2013) and after (2014–2015) the onset of the marine heatwave.

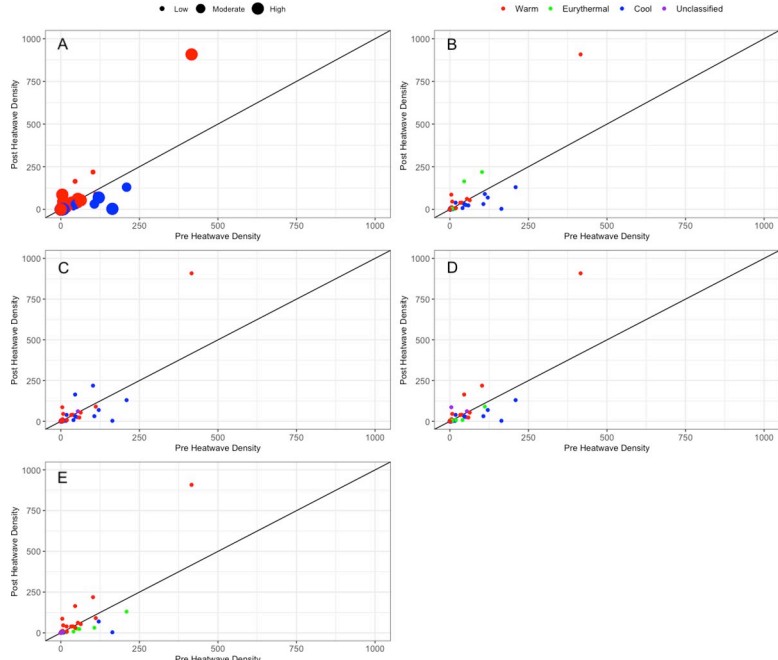

**Fig 2. Variations in species response by classification metric.** Scatterplots of individual species responses to the marine heatwave are shown with pre-heatwave (2012–2013 summed density) on the x axis and post-heatwave (2014–2015 summed density) on the y axis. Points that fall below the black line are species that responded with density declines while points above the line experience density increases in response to the heatwave. Points are colored according to their classification (red for warm, green for eurythermal, blue for cool and purple for unclassified) for the A) composite classification B) expert opinion C) range limits D) museum Collections and E) diver surveys. In the composite classification, point size enlarge as agreement among classification data rises.

driven change to fish community composition appears to be heavily driven by the ten most responsive species. Fortunately, the composite scores for the majority of these top ten species were in high agreement and we can be confident in the utility of the thermal classification scheme as a tool for accurately tracking community-level response. As research and management aims to rapidly detect and respond to climate-driven ecological changes, users of classification schemes must have confidence that each response group is constructed such that all members respond similarly to a climate driver, but is inclusive of the range of intensity of responses species can display. Further work describing ecosystem response to the marine heatwave in this region can be found in Freedman et al. 2020 [19]. In this study, the number of kelp forest fish species classified as having warm and cool affinities differed by the data source used for classification. Differences in temporal and spatial resolution for each data source likely contributed to the number of species in each classification group and the number left unclassified due to data gaps.

The number of species left unclassified (e.g., present in the ecosystem but not captured in the classification data source) was highest (n = 61) for the subtidal diver survey data source. Recall that we used only two years (2009–2010) of the full SCUBA survey data for the classification in part to achieve complete spatial coverage of California, and also to avoid circularity in using the same data for classification as we are using to test the classification. Thus, 61 species were observed in the complete dataset that were not observed in 2009 or 2010. Researchers wanting to use *in situ*, field surveys in their composite classifications might have some issues because long running continuous surveys are uncommon across the world and are rarely conducted across large spatial domains [20]. The other two spatial datasets, museum collections and range limits, contained data from longer time periods and had fewer species left unclassified.

Museum collection records have spatial and temporal biases as well that may limit their effectiveness as a data source to determine thermal affinity when used in isolation. While Museum collections may encompass very long time periods, effort by collectors likely are infrequent, inconsistent and may not span the entire range of an individual species. This means collection events are likely affected by sampling bias and may lead to incorrect classifications alone. However, incorporating sampling events over a long time period allows classifications to incorporate historic variability in ecosystem structure.

Data sources can be biased in other ways that users of classification schemes should consider when implementing classification schemes on their data. Caution should be used when community indices are based on a single source of information as they are potentially error prone. For example, in the California Current Large Marine Ecosystem, El Niño events cause pulses of species recruitment northward of their typical ranges [21, 22] where they might establish but in very low densities; thus, biasing range limits as a data metric. Used alone, this could cause warm- affiliated species to be mis-classified as eurythermal or cool. In fact, in this study, range limits had the highest proportion of species classified as cool. Misclassification of some warm species may also explain why the range limits-based classification method was the only method that had cool water species increasing in abundance in response to the marine heat wave. Marine heatwaves, which have occurred locally [23, 24], also could drive warm species range expansion and alter classification findings [25]. Using older data prior to the prevalence of these phenomena will help establish community change from a relevant baseline in the Anthropocene.

Given that biases exist are likely to exist in any source of data typically used to classify thermal affinity, using a composite classification scheme is one option to mitigate bias. Using only species densities is often limited by sampling design and the potential for "double dipping"– that is, using the same dataset to classify species as to track them. Although we included

density datasets in our classification and went on to use this classification to assess trends through time, using additional classification methodology in equal weights and a temporally limited sample allows us more confidence in testing the scheme on density trends. Including Expert Opinions into the composite score also allows for potential data biases to be cross-checked with the understanding of experts in the field. In this study, most species had high agreement (n = 78) between datasets, however a large number of species (n = 47) had low agreement between techniques. This means that these 47 species had a high likelihood of being classified differently depending on which dataset was used. High levels of species misclassifications would limit the scheme's utility and could lead to incorrect conclusions about true community responses to climate change [20, 26, 27]. By using the majority classification across all datasets, a composite score limits the potential a species is misclassified by spreading misclassification risk across multiple sources of data.

In our study, the composite classification method resulted in density responses of warm and cool species that were less extreme compared to single data source techniques and as expected, did not have any of the maximum or minimum community responses to the marine heatwave. For most individual data sources, warm water species increased and cool water species decreased in response to the marine heatwave [25, 28]; however the magnitude of change was variable and using only Range Limits as a data source, we found the opposite pattern, that cool water species increased in abundance. False or inflated extreme responses to acute climate drivers could cause overcompensation with conservation measures [29–31]. By using the composite classification, there is a risk reduction of lost resource use opportunities, potential undesired ecological outcomes or potentially not intervening quickly enough to limit environmental impacts.

While we used a composite score based on agreement between four data sources, other means of averaging could be used as well as differential weighting schemes (we weighted all four data sources equally). For example, if one data source was deemed of higher 'quality' than another, it could be weighted more highly in the creation of a composite. Here we provide a simple example of a composite thermal indicator that uses data from multiple sources and that captured changes in fish communities over a thermal event in our region.

## Conclusion

Classification schemes offer a quick, easy to interpret, and functional way to quantify community responses to climate change. Commonly used single trend measures, such as Community Thermal Index (CTI), can mask ecological complexity and make it difficult to make alternate management decisions for groups that respond differently to climate change [4, 8]. Because CTI is a single number metric, further analysis will always be warranted, to understand the actual structural changes driving CTI shifts. By splitting species into groups that likely will have differential responses to climate drivers, conservation practioners can create finer resolution conservation measures and attempt to mitigate direct pressures on each group independently (Fig 3). In contrast, traditional ecological community analysis metrics (e.g., PCA, CCA, regression trees) may be too complex for managers to make actionable conservation decisions as they can create too many groups that may confound responses to drivers of interest or not be amenable to policy or management actions [11, 17].

There needs to be flexibility to increase, maintain and relax management strategies where appropriate and single number ecological metrics are not able to provide adequate information to inform complex management strategies [32–34]. Managers could mix restoration efforts [35, 36], species removal [37–39], fisheries management [34], and other management methods to mitigate changes in ecosystem structure. However, careful thought is required in

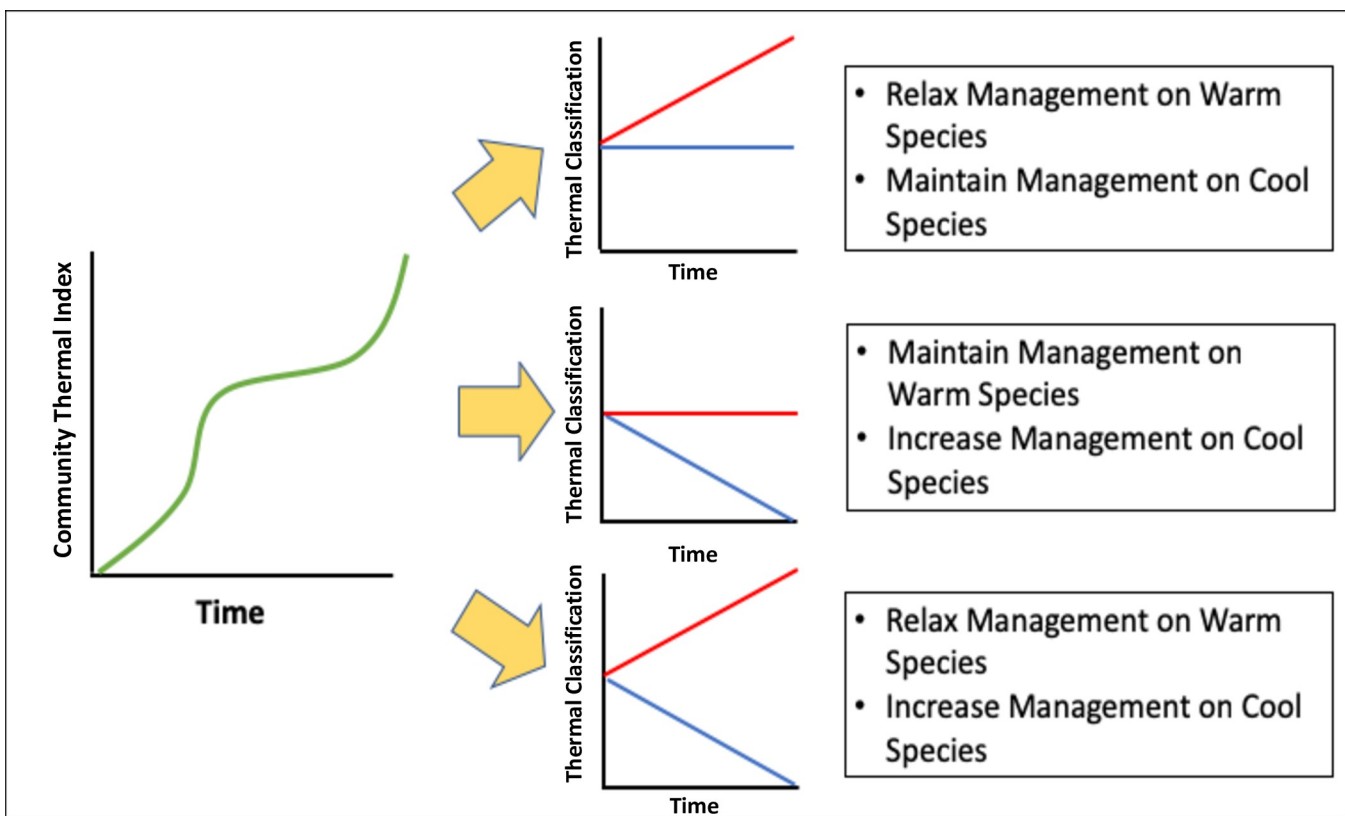

**Fig 3. Comparisons between CTI and our multi-data source, thermal classification scheme.** The single trend line for Community Thermal Index (CTI) obscures the responses of specific groups of species to environmental drivers challenging management decision-making. The Y-axis on each of these graphs would represent some indicator of ecosystem change such as density or biomass. Because CTI is a single trend line, it masks the potentially different contributions of species or species groups to the trend. For example, the three scenarios on the right could all result from the single CTI trend shown on the left. The additional resolution offered by thermal classification gives managers finer scale information to enact specific actions that either benefit or boost populations at risk (e.g., decreased fisheries take, restoration action, marine protected) or increase resource use on populations experiencing increases (e.g., increased fisheries take, species culling).

the creation and use of classification schemes to ensure findings are consistent with known ecological phenomena [8, 17]. Combining sources of data to minimize temporal and spatial limitations is one way to address potential biases that could potentially misinform important management decision and lead to unintended outcomes. As managers struggle to keep up with rapidly increasing temperatures and more frequent marine heat waves, rapid assessment of local ecology with accurate classification strategies will be valuable to maintain critical ecosystem function and design effective conservation measures.

## Supporting information

**S1 Fig. A map of the study area and relevant biogeographic boundaries.** This map depicts the biogeographic break of Point Conception (shown in purple) in relation to the study area (bounded by the red box) where data was collected to assess ecosystem response to the marine heatwave.
(TIF)

**S1 Table. The species classification methods.** The classifications of species (n = 152) are shown below across data sources and their resulting composite classification and agreement ranking. Species below the second black line are found in the recruitment dataset but not the

diver survey data. Blank spaces denote that data was not available to create a classification and species are unclassified.

(DOCX)

## Acknowledgments

This study relies heavily on the continuous work of the Partnership for studies of Coastal Oceans (PISCO) and Vantuna Research Group (VRG) dive teams. Fishbase and Vertnet and their contributors also provided key datasets. Participation in our expert opinion poll was vital and we thank everyone who participated and volunteered their time classifying fishes.

## Author Contributions

**Conceptualization:** R. Freedman, J. A. Brown, J. E. Caselle.

**Data curation:** R. Freedman, J. A. Brown, C. Caldow, J. E. Caselle.

**Formal analysis:** R. Freedman, J. A. Brown, J. E. Caselle.

**Funding acquisition:** R. Freedman, C. Caldow.

**Investigation:** R. Freedman, J. A. Brown, C. Caldow, J. E. Caselle.

**Methodology:** R. Freedman, J. A. Brown, C. Caldow, J. E. Caselle.

**Project administration:** R. Freedman, C. Caldow, J. E. Caselle.

**Resources:** R. Freedman, C. Caldow, J. E. Caselle.

**Software:** R. Freedman, C. Caldow.

**Supervision:** R. Freedman, J. A. Brown, C. Caldow.

**Validation:** R. Freedman, J. A. Brown, J. E. Caselle.

**Visualization:** R. Freedman, J. A. Brown, J. E. Caselle.

**Writing – original draft:** R. Freedman, J. A. Brown, C. Caldow, J. E. Caselle.

**Writing – review & editing:** R. Freedman, J. A. Brown, C. Caldow, J. E. Caselle.

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
