## [Decision Letter · Decision Letter 0]

11 Mar 2021

PONE-D-20-36102

Species-specific thermal classification schemes can improve climate related marine resource decisions

PLOS ONE

Dear Dr. Freedman,

Thank you for submitting your manuscript to PLOS ONE. After careful consideration, we feel that it has merit but does not fully meet PLOS ONE’s publication criteria as it currently stands. Therefore, we invite you to submit a revised version of the manuscript that addresses the points raised during the review process.

I believe that you will be able to address in your revision the useful and constructive comments and suggestions made by reviewer #1, and include further explanations especially on the thermal width of the species considered in your study as well as consideration about lags in the response to the heatwave in relation to recruitment processes.

We look forward to receiving your revised manuscript.

Kind regards,

Andrea Belgrano, Ph.D.

Academic Editor

PLOS ONE

Journal Requirements:

2. Thank you for stating the following in the Competing Interest section:

We note that one or more of the authors are employed by a commercial company: ECOS Consulting, LLC

(2) Please also provide an updated Competing Interests Statement declaring this commercial affiliation along with any other relevant declarations relating to employment, consultancy, patents, products in development, or marketed products, etc.  

Please respond by return email with an updated Funding Statement and Competing Interests Statement and we will change the online submission form on your behalf.

Reviewers' comments:

Reviewer's Responses to Questions

**Comments to the Author**

1. Is the manuscript technically sound, and do the data support the conclusions?

Reviewer #1: Partly

2. Has the statistical analysis been performed appropriately and rigorously? 

Reviewer #1: No

3. Have the authors made all data underlying the findings in their manuscript fully available?

Reviewer #1: No

4. Is the manuscript presented in an intelligible fashion and written in standard English?

Reviewer #1: Yes

5. Review Comments to the Author

Reviewer #1: Dear authors,

I enjoyed reading “Species-specific thermal classification schemes can improve climate related marine resource decisions”. I personally find predictions about the effects of warming based on species being cold or warm adapted vague, partly because classifications do not seem robust and often seem based on simplistic views of a species. This is a neat paper that address just that uncertainty, by developing a classification scheme for the thermal affinity of fish using multiple data sources (including expert opinion) in order to robustly classify species as either “warm“ or “cold”. Next, differences in the response of these two thermal groups to a marine heatwave, acting as a natural experiment, are evaluated. I have a few major and some minor comments that I hope the authors find useful.

Major comments

- Line 84: I was wondering if you had discussed the observation that that warm-adapted/tropical species have narrower thermal tolerances that sup-tropical or temperature species. This would suggest that cold/temperate species might actually be better to withstand heatwaves because of their wider thermal windows than warm/sub-tropical/tropical species who live closer to the “edge”. In other words, what matters for a species ability to survive a heatwave is more linked to the thermal width rather than at which temperature the preferred temperature is. Is this something that also applies to your species, or would you say that all your species have more or less the same thermal width?

- A follow up question related to the natural experiment that is the heatwave: is the density a good metric in this case? I think this depends on what drives the fluctuations in abundance of these species in this system. You only use two years (2012-2013 and 2014-2015). The densities in the heatwave years are likely very dependent on the recruitment in the normal years. Basically, what is driving the increase in density of the warm species in the warm period? Larger recruitments? If so, are fish recruited in the two-year period sampled by the survey in the same period, or are they too small? Or is that warm species are moving into the area because it’s warmer? Do you expect a delayed response in density from a heatwave? (Same goes for the difference in cold species density but opposite). Essentially, could you explain a little bit what the relevant mechanisms are?

- Did the heatwave act uniformly across the whole study area? Because you use density below and above Point Conception to characterize species in one of the methods, I guess it’s important to state explicitly that that the heatwave acted equally above and below Point Conception, else it could be a confounding effect.

- Line 182: The approach to evaluate the responses seems too simplistic and descriptive. I see that the average changes are very clear (by thermal classification), but nothing about the uncertainty. Have you considered fitting a regression to this? For instance, for each species, evaluate if there’s a difference in density between cold and warm years, and then see if there’s an effect of the thermal classification (or set it up in some other way). Because with all species pooled, I can’t tell if the large changes are driven by a few species or a little bit by all species in groups. Or even if or how many of the warm species show declines in densities in the heatwave, and vice versa. Also, it would be interesting to see if the response was clearer for species with better agreement across classification methods (higher rank). Without these analyses, it’s very difficult to assess how statistically clear the findings are.

- Line 141: I wonder if it’s interesting to actually look at the eurythermal species, kind of as a control. You would expect them to be somewhere in between the cold and warm species in terms of changes in density, right?

Minor comments

- Line 44: “natural resources”

- Line 127: A map of the area would have been nice for readers who are not familiar.

- Line 157: Where the experts instructed to not google species or use FishBase? Otherwise, data such as thermal ranges would be easy to look up. Where the classification from source 3 and 4 more similar, for instance?

- Figure 1: see major point about the analysis. I think it would be great to analyze this differently to get a feeling for the statistical uncertainty and the distributions of species responses.

- Figure 2: needs to be explained more. I think it’s best to spell out what CTI stands for here since it’s only mentioned once in the text (and there it can be explained, not just mentioned). I don’t really understand what the y-axis represents in the right graphs. What is meant by management here? Is it something that benefits or aims to protect the population here, or relaxed management as in fewer rules leading to higher exploitation? And how is it related to the left graph?

6. PLOS authors have the option to publish the peer review history of their article (what does this mean?). If published, this will include your full peer review and any attached files.

Reviewer #1: No

---

## [Author Response · Author response to Decision Letter 0]

29 Mar 2021

Major comments

Line 84: I was wondering if you had discussed the observation that that warm-adapted/tropical species have narrower thermal tolerances that sup-tropical or temperature species. This would suggest that cold/temperate species might actually be better to withstand heatwaves because of their wider thermal windows than warm/sub-tropical/tropical species who live closer to the “edge”. In other words, what matters for a species ability to survive a heatwave is more linked to the thermal width rather than at which temperature the preferred temperature is. Is this something that also applies to your species, or would you say that all your species have more or less the same thermal width?

This is a very interesting point and while we cannot know the thermal widths of our species, we have added mention of this in several places. A number of species traits will impact species range shifts due to climate change including thermal width, maximum survivable temperature, preferred temperature and the plasticity around these traits. In our marine transition system, we would expect that both cool water and warm water species are most likely at the edge of their thermal width in our study site. We added text to that effect in lines 95-100: “Given that this area is a marine transition zone, we expect that fish species will be at the edge of their physiological thermal width and may be able to quickly respond to climate impacts”. Actual physiological data to ascertain the thermal width of each species is rare and we can’t be sure whether cool water or warm water species in this system tend to have larger thermal widths. We have also added some text to our introduction to bring up this point: Line 57-58: Because physiological data on most species’ thermal tolerances is absent, the most commonly used classifications are based on species thermal tolerances as inferred from biogeography in situ. Our manuscript is focused on addressing these composite classification scores as a tool and we believe our classification scheme captures the variation of species responses based on variability of thermal width. We have added some additional text in the discussion on this point in lines 284-313: Species responses to temperature shifts will vary both on temperature tolerance width, maximum survivable temperature, preferred temperature and the plasticity around these traits (4,19). In our system, thermal driven change to fish community composition appears to be heavily driven by the ten most responsive species. Fortunately, the composite scores for the majority of these top ten species were in high agreement and we can be confident in the utility of the thermal classification scheme as a tool for accurately tracking community-level response. As research and management aims to rapidly detect and respond to climate-driven ecological changes, users of classification schemes must have confidence that each response group is constructed such that all members respond similarly to a climate driver, but is inclusive of the range of intensity of responses species can display.

- A follow up question related to the natural experiment that is the heatwave: is the density a good metric in this case? I think this depends on what drives the fluctuations in abundance of these species in this system. You only use two years (2012-2013 and 2014-2015). The densities in the heatwave years are likely very dependent on the recruitment in the normal years. Basically, what is driving the increase in density of the warm species in the warm period? Larger recruitments? If so, are fish recruited in the two-year period sampled by the survey in the same period, or are they too small? Or is that warm species are moving into the area because it’s warmer? Do you expect a delayed response in density from a heatwave? (Same goes for the difference in cold species density but opposite). Essentially, could you explain a little bit what the relevant mechanisms are?

Again, a great comment and unfortunately, again, we are not totally sure of the exact mechanism of the increase although we are working on other analyses and papers that explore this further. Basically, we selected density as our metric to test our thermal classification because it does potentially account for all the mechanisms you highlight. We have added some text to better describe this in Lines 218-221: Density metrics were selected to test the thermal classification for any response to the heatwave because density measures will be inclusive of a number population and demographic responses including larval recruitment, increased survivorship, adult migration and increased habitat use.

- Did the heatwave act uniformly across the whole study area? Because you use density below and above Point Conception to characterize species in one of the methods, I guess it’s important to state explicitly that that the heatwave acted equally above and below Point Conception, else it could be a confounding effect.

The 2014 heatwave’s impact was extensive and impacted the entire US West Coast both above and below Point Conception. However, densities used in the classification scheme (2009-2010) were from exclusively before the heatwave appeared and therefore should not be impacted from the heatwave. We add some text to lines 138-141 to further clarify this for readers: While surveys were conducted as part of the same monitoring program (i.e. PISCO), this larger scale snapshot does not overlap the timeframe of the data used for the community change analysis described below and therefore should not be influenced by the 2014 marine heatwave or lead to redundancies in the analysis. 

- Line 182: The approach to evaluate the responses seems too simplistic and descriptive. I see that the average changes are very clear (by thermal classification), but nothing about the uncertainty. Have you considered fitting a regression to this? For instance, for each species, evaluate if there’s a difference in density between cold and warm years, and then see if there’s an effect of the thermal classification (or set it up in some other way). Because with all species pooled, I can’t tell if the large changes are driven by a few species or a little bit by all species in groups. Or even if or how many of the warm species show declines in densities in the heatwave, and vice versa. Also, it would be interesting to see if the response was clearer for species with better agreement across classification methods (higher rank). Without these analyses, it’s very difficult to assess how statistically clear the findings are.

The goal of this manuscript is to do basic testing of our classification scheme and see if each thermal grouping responds as a whole. Evaluations and assessments are meant to be simple by design, so they can be easily interpreted and understood by managers and non-scientific stakeholders. Our pooled approach is geared towards managers that want to be able to track changes in overall community composition. The goal of this paper is to show the utility and strength of the thermal classification technique and further exploration of the heatwave’s effects on the ecology of the region can be found in Freedman et al 2020.

In order to address this comment and show readers the variation of responses by species, we are adding an additional graph of pre-heatwave and post-heatwave species responses as supplementary materials. Readers will be able to see the variation in pre- and post-heatwave densities in the region by species. Additional text discussing this aspect are interspersed in the methods, results and discussion. In methods, description of these graphs Lines 216-218: Pre- and post-heatwave densities of individual species were plotted against one another, for each classification metric (including the composite metric) separately, and qualitatively described.

In the results, description of variability and spread of response is found in Lines 253-267 In the ten species that had the largest differences pre and post heatwave (Figure 2), six were of high agreement, two were moderate agreement and two were low agreement (Fig 2). These strongest species responses were split pretty evenly between warm (n=6) and cool species (n=6). The summed changes in fish density responses to the heatwave were heavily driven by these top 10 species which was 6 times total observed change in density for the remaining species (Fig 2).

 In the discussion, we further describe these finding in lines 286-313: In our system, thermal driven change to fish community composition appears to be heavily driven by the ten most responsive species. Fortunately, the composite scores for the majority of these top ten species were in high agreement and we can be confident in the utility of the thermal classification scheme as a tool for accurately tracking community-level response. As research and management aims to rapidly detect and respond to climate-driven ecological changes, users of classification schemes must have confidence that each response group is constructed such that all members respond similarly to a climate driver, but is inclusive of the range of intensity of responses species can display. Further work describing ecosystem response to the marine heatwave in this region can be found in Freedman et al. 2020.

- Line 141: I wonder if it’s interesting to actually look at the eurythermal species, kind of as a control. You would expect them to be somewhere in between the cold and warm species in terms of changes in density, right?

Because eurythermal classifications were reached both through direct classifications as euthythermal or through splitting of classification metrics, we do not feel it would be appropriate to use this group as a control.

Minor comments

- Line 44: “natural resources”

This grammatical fix has been changed in the text

- Line 127: A map of the area would have been nice for readers who are not familiar.

A map figure has been added as a supplementary figure to orient readers who may not be familiar with the region.

- Line 157: Where the experts instructed to not google species or use FishBase? Otherwise, data such as thermal ranges would be easy to look up. Where the classification from source 3 and 4 more similar, for instance?

No such instruction was given to experts but sums of species by classifications show that these classification criteria methods were the most dissimilar for warm and eurythermal species which would imply that experts did not look up species classifications. We added this text to the manuscript to address this gap in Lines 180-185: While we did not explicitly tell experts to not use tools, such as FishBase, to identify species ranges, - they were encouraged to rely on their natural history insight. Post hoc comparisons suggest it is unlikely that experts gathered information from other sources; the number of cool and eurythermal species classified by expert opinion and ranges from Fishbase were the two most dissimilar.

- Figure 1: see major point about the analysis. I think it would be great to analyze this differently to get a feeling for the statistical uncertainty and the distributions of species responses.

A new figure (Figure 2) has been added so readers can see the statistical uncertainty and distribution of species responses.

- Figure 2: needs to be explained more. I think it’s best to spell out what CTI stands for here since it’s only mentioned once in the text (and there it can be explained, not just mentioned). I don’t really understand what the y-axis represents in the right graphs. What is meant by management here? Is it something that benefits or aims to protect the population here, or relaxed management as in fewer rules leading to higher exploitation? And how is it related to the left graph?

The addition of the new Fig 2 means that old Fig 2 is now Fig 3. For this figure, CTI has been changed to Community Thermal index in the figure and more text has been added to the caption to explain the questions raised by the reviewer. The caption for what is now Figure 3 reads as: The single trend line for Community Thermal Index (CTI) obscures the responses of specific groups of species to environmental drivers challenging management decision-making. The Y-axis on each of these graphs would represent some indicator of ecosystem change such as density or biomass. Because CTI is a single trend line, it masks the potentially different contributions of species or species groups to the trend. For example, the three scenarios on the right could all result from the single CTI trend shown on the left. The additional resolution offered by thermal classification gives managers finer scale information to enact specific actions that either benefit or boost populations at risk (e.g., decreased fisheries take, restoration action, marine protected) or increase resource use on populations experiencing increases (e.g., increased fisheries take, species culling).

---

## [Editor Report · Decision Letter 1]

14 Apr 2021

Species-specific thermal classification schemes can improve climate related marine resource decisions

PONE-D-20-36102R1

Dear Dr. Freedman,

We’re pleased to inform you that your manuscript has been judged scientifically suitable for publication and will be formally accepted for publication once it meets all outstanding technical requirements.

Kind regards,

Andrea Belgrano, Ph.D.

Academic Editor

PLOS ONE

Additional Editor Comments (optional):

The revised manuscript addresses all the suggestions/comments made during the first round of reviews, thank you.

---

## [Editor Report · Acceptance letter]

19 Apr 2021

PONE-D-20-36102R1 

Species-specific thermal classification schemes can improve climate related marine resource decisions 

Dear Dr. Freedman:

I'm pleased to inform you that your manuscript has been deemed suitable for publication in PLOS ONE. Congratulations! Your manuscript is now with our production department. 

Kind regards, 

on behalf of

Dr. Andrea Belgrano 

Academic Editor

PLOS ONE